# Effect of Musts Oxygenation at Various Stages of Cider Production on Oenological Parameters, Antioxidant Activity, and Profile of Volatile Cider Compounds

**DOI:** 10.3390/biom10060890

**Published:** 2020-06-10

**Authors:** Tomasz Tarko, Aleksandra Duda-Chodak, Paweł Sroka, Magdalena Januszek

**Affiliations:** Department of Fermentation Technology and Microbiology, Faculty of Food Technology, University of Agriculture in Krakow, ul. Balicka 122, 30-149 Krakow, Poland; a.duda-chodak@ur.krakow.pl (A.D.-C.); p.sroka@ur.krakow.pl (P.S.); magdalena.kostrz@urk.edu.pl (M.J.)

**Keywords:** micro-oxygenation, cider, antioxidant activity, volatile compounds

## Abstract

The micro-oxygenation of musts may affect the quality of a finished alcoholic beverage. The aim of this study was to determine the effect of micro-oxygenation at various stages of fermentation on oenological parameters, antioxidant activity, total polyphenol content, and profile of volatile cider compounds fermented with various yeast strains. Rubin cultivar must was inoculated with wine yeast, cider yeast, distillery yeast, and wild yeast strains. Some of the inoculated samples were oxygenated immediately after yeast inoculation, and some on the second and third fermentation days. The control sample was non-oxygenated must fermented in bottles. Higher extract concentration and acidity as well as lower potency were observed in cider treated with micro-oxygenation. Must oxygenation in most cases contributed to the reduction of polyphenol content and to the antioxidant activity of ciders, especially when fermented using wild yeast. The oxygenation of musts before fermentation caused an increase in the content of esters and alcohols in ciders. However, the oxygenation of musts during fermentation reduced the concentration of these volatile components. The oxygenation of musts during fermentation produced a differentiated effect on terpenoid concentration in ciders.

## 1. Introduction

Alcohol fermentation plays a major role in the production of wines and ciders. It is an anaerobic process, therefore excessive amounts of oxygen can stop the fermentation. Contact with oxygen is usually considered to be a negative factor affecting the final quality of wines and ciders. Polyphenolic compounds, which are precursors of many compounds responsible for the color, taste, and aroma of wines and ciders, react with oxygen and get degraded, which adversely affects the sensory properties of wines, especially white wines. In the presence of oxygen, polymerization of polyphenolic compounds occurs, which leads to the formation of dark-colored components (from yellow to brown). The consequence of this process is the darkening of young wines, which is unacceptable for white wines. Exposure of musts for white wine to oxygen has a destructive effect on wine’s aroma, as the compounds responsible for the fresh, fruity aroma of wine are easily oxidized [1,2]. Despite the greater buffering capacity of musts from red grape varieties in relation to oxygen, even they can undergo adverse changes upon contact with oxygen. High oxygen concentrations can cause oxidation of polyphenolic compounds and their precipitation in the form of sediment. During ripening, polyphenolic compounds play a huge role in creating color and appropriate taste sensations (fullness and astringency). Oxidation of polyphenolic compounds reduces their concentration in the final product and, as a consequence, negatively affects its quality [3]. 

However, it is necessary to stress the need to add small amounts of oxygen to the must before fermentation, as this contributes to the proper cell structure of yeast. Yeast cells need oxygen to produce sterols (mainly ergosterols) and unsaturated fatty acids, which play an important role in the fluidity and activity of membrane-associated enzymes that influence ethanol tolerance, fermentative capability, and the viability of yeast [2,4]. In addition, the use of an appropriate dose of oxygen can reduce the formation of sulfur compounds and shorten the fermentation time, which is probably associated with a greater resistance of yeast cells to ethanol [1,5]. Oxidation of must also affects the synthesis of volatile compounds by yeasts. Esters (including acetates and ethyl esters), higher alcohols, medium-chain fatty acids, aldehydes, or ketones formed during the fermentation stage are responsible for creating the color and aroma of wines [6,7].

Micro-oxygenation accelerates wine maturation and is most often used for wines with a high tannin content. Oxygen can modify the sensory properties of wine (taste, smell, and color), which is primarily due to its strong oxidizing properties. Terpenes, polyphenols, and other compounds that are constituents of wine are susceptible to oxidation reactions, which consequently affects a drink’s composition [2,8,9,10].

The aim of this study was to determine the effect of micro-oxygenation at various stages of fermentation on oenological parameters, antioxidant activity, total polyphenol content, and profile of volatile cider compounds fermented with various yeast strains.

## 2. Materials and Methods 

### 2.1. Ciders Preparation

The apples (Rubin cultivar) used in these experiments were obtained from a pomological orchard of the University of Agriculture, located in Garlica Murowana (near Krakow, Poland). The apples were washed, ground, and pressed on a Zottel hydraulic press (35 L). The musts (0.5 L) obtained from particular cultivars were poured into bottles (0.7 L). The musts were then inoculated with various yeast strains (Erbslöh, Geisenheim, Germany): wine (Elegance), cider (Gozdawa), distillery (Ethanol RED), or wild (Wild & Pure) at 0.2 g/L (dry yeasts were hydrated according to the manufacturer’s recommendations). One part of the samples was oxygenated through in-system sterile air barbotage to the bottles: 10 s oxygenation followed by a 10 s interval, for 1 h (air flow 1.6 L/min, this stage is referred to as must oxygenation). The bottles with the inoculated must were closed with glycerin-filled fermentation tubes and then allowed to ferment for two weeks at 20 °C. Other samples were oxygenated as above on the second and third day of fermentation (oxygenation during fermentation), while the control samples consisted of non-oxygenated musts fermented in bottles. The ciders were drained from the yeast sludge and aged at 4 °C for three weeks. All experiments were performed in triplicate.

### 2.2. Determination of Total Acidity, Volatile Acidity, Total Extract Content, and Ethyl Alcohol Content

The determination of total extract content, ethyl alcohol content, total acidity, and volatile acidity was conducted in accordance with the methods recommended by the International Organisation of Vine and Wine [11]. Total extract content and ethanol content in ciders were determined by distillation methods using pycnometric density determination, total acidity was determined with the potentiometric method, and volatile acidity was determined using the titration method. All experiments were performed in triplicate.

### 2.3. Free Amino Nitrogen (FAN) Content

Free amino nitrogen (FAN) was determined with the ninhydrin method. The absorbance of the samples was measured at a wavelength λ = 575 nm [12]. All experiments were performed in triplicate.

### 2.4. Antioxidant Activity

The antioxidant capacity of the samples was determined by the diammonium salt of 2,2′-azino-bis (3-ethylbenzothiazoline-6-sulfonic) acid (ABTS) cation radical scavenging assay [13]. Active ABTS was produced through a chemical reaction with potassium persulfate. The absorbance was measured spectrophotometrically at 734 nm. The results obtained were compared with the those obtained for Trolox (synthetic vitamin E) and expressed as mg Trolox/100 mL (TEAC, Trolox Equivalent Antioxidant Capacity). Absorbance measurements were performed on a UV–VIS Beckman spectrophotometer (type DU 650). All experiments were performed in triplicate.

### 2.5. Total Polyphenol Content

The total polyphenol content was determined with a spectrophotometric method (UV–VIS Beckman spectrophotometer) with Folin–Ciocalteu reagent [13]. The results of the total polyphenols were expressed as catechin equivalent (mg/100 mL), based on a standard curve. All experiments were performed in triplicate.

### 2.6. Analysis of Polyphenolic Compound Profiles

Prior to analysis, the samples were filtered through a nylon syringe filter (0.45 μm, Chemland, Stargard Szczecinski, Poland). The analysis of the polyphenol profiles was carried out using a high-performance liquid chromatographer (HPLC, Shimadzu, Japan) equipped with a DAD detector. The Synergi Fusion RP-80A column 150 mm × 4.6 mm (4 μm) (Phenomenex, Torrance, California, USA), thermostated at 30 °C, was used for all analyses. Acetonitrile (POCh Gliwice) and a 2.5% aqueous solution of acetic acid (POCh) were used as a mobile phase. The gradient program and detection wavelengths are described in detail by Tarko et al. [14].

For quantitative analyses, calibration curves were prepared for the following standards: ferulic acid, caffeic acid, chlorogenic acid, gallic acid, hippuric acid, *p*-coumaric acid, protocatechuic acid, ellagic acid, (+)-catechin, quercetin, resveratrol, kaempferol (Sigma Aldrich), phloridzin, (-)-epicatechin, procyanidins B1 and B2, cyanidin-3-*O*-galactoside, cyanidin-3-*O*-sambubioside, cyanidin-3-*O*-arabinoside, cyanidin-3-*O*-glucoside, delphinidin-3-*O*-glucoside, quercetin-3-*O*-rutinoside, quercetin-3-*O*-glucoside, pelargonidin-3-*O*-glucoside, and peonidin-3-glucoside (Extrasynthese, Genay, France). Compounds were identified by comparing the retention time of individual peaks and UV spectra with those of the standards (using the spectrum library for the standards). Polyphenols not detected in any experimental variants were not included in the tables. All experiments were performed in triplicate.

### 2.7. Analysis of Volatile Compounds

The volatile compounds were determined using a modified Grützman et al. [15] method. A sample of 2 mL was placed into a 15 mL headspace vial, and 50 μL of an internal standard solution (5 mg/L of ethyl nonanoate) was added. Then a solid-phase microextraction (SPME) fiber (85 μm Carboxen Polydimethylsiloxane, Supelco, St. Louis, MI, USA) was placed in the headspace above the sample, and the vial was incubated for 30 minutes at 40 °C. The fiber was subjected to thermal desorption in a gas chromatographic injector at 250 °C.

The chromatographic separation was carried out on a Clarus 580 apparatus equipped with a flame ionization detector (FID) (PerkinElmer, Walthman, MA, USA) and a Crossbond dimethylpolysiloxane column (60 m in length, 0.25 mm of inner diameter, 1.4 μm of film thickness, Restek, Bellefonte, PA, USA). The carrier gas flow (He) was 2 mL/min, and the temperature program was: 35 °C, 6 min; 8 °C/min up to 180 °C; 12 °C/min up to 220 °C; 25 min. The inlet and FID temperature was 250 °C. An HT2800T autosampler (HTA Brescia, Italy) was used, and PerkinElmer Total Chrom 6.3.2 software (PerkinElmer, Walthman, MA, USA) was used to integrate the results. Identification was performed based on the retention times compared with those of the standards for the volatile compounds, and quantification was carried out by internal standardization, using 4-methyl-2-pentanol (alcohols) and ethyl nonanoate (esters) as the internal standards. The standard solutions were prepared in synthetic wine (20 g/L sucrose, 5 g/L tartaric acid, 10% ethanol). All experiments were performed in triplicate.

### 2.8. Analysis of Terpenoids

A 40 mL sample was measured into a test tube (50 mL); 100 μL of internal standard solution (5 mg/L anethol) and 4 mL of hexane were added, and extraction was performed on a rotary shaker (350 cpm, amplitude 3, 1 h). Then, the hexane layer was removed, transferred to the equation vessels, and centrifuged (2154× *g*, 10 min). The hexane layer was collected, and the chromatographic analysis was performed.

The separation was carried out using an HP 5820 camera, a Stabilwax column (30 m, 0.25 mm) and a FID. The carrier gas flow (He) was 2 mL/min, and the temperature program was: 35 °C for 1 min, increased by 4 °C/min up to 250 °C; then held at 250 °C for 5 min. The detector and dispenser temperature was 250°C. An HT3000A autosampler (from HTA company, Brescia, Italy) was used, and the results were integrated using Clarity 7.2 software from DataApex Ltd. All experiments were performed in triplicate.

The limit of detection (LOD) for the analyzed terpenoids was 0.003 mg/L, and the limit of quantification (LOQ) was 0.01 mg/L.

### 2.9. Statistical Analysis

There were at least three physical repetitions of each experimental setting. All samples were analyzed once, but in the case of discrepancies in the results, the analysis was repeated. Results are shown as the arithmetic mean with standard deviation (± SD). The statistical analysis was performed using InStat v.3.01 (GraphPad Software Inc., USA). A single-factor analysis of variance (ANOVA) with post-hoc Tukey’s test was applied to determine the significance of differences. The Kolmogorov–Smirnov test was carried out to assess the normality of distribution. 

## 3. Results and Discussion

### 3.1. The Effect of Oxygenation on Oenological Parameters of Ciders

Alcohol concentration depends primarily on the amount of fermentable sugars in the must and on the type of yeast. Ciders can contain from 1.2% to 8.5% of alcohol. The obtained results were in this range, as expected. The must extract before fermentation was about 130 g/L, and its content in ciders ranged from 16.43 to 18.8 g/L. The loss of extract (the amount of fermented sugars) was consistent with the amount of alcohol produced by the yeast and depended on the type of yeast used.

Based on the results obtained, it can be concluded that the use of oxygenation significantly reduced the concentration of ethanol in the tested samples (by about 10–15% compared to the control samples) (Table 1). Similar results were presented by Sirén et al. [16], showing a reduction in the concentration of ethanol in oxygenated wines, compared to controls. The lower concentration of ethanol in cider may be associated with oxidative stress of yeast during fermentation, resulting from oxygenation. Oxidative stress is one of the main causes of early death of yeasts during the fermentation process [1]. Furthermore, the reduction in ethanol concentration may be due to its oxidation to acetaldehyde [17]. During oxidation, the formation of acetaldehyde from ethanol would promote the formation of ethyl bridges between flavanols and between flavanols and anthocyanins [18]. The decrease in ethanol concentration may also be caused by the increased use of sugar sources for biomass growth [19]. Consequently, reducing the concentration of substrate in the must results in a decrease in the ethanol content of fermented apple drinks subjected to micro-oxygenation. On the other hand, a properly selected dose of oxygen added to the must before fermentation and also at the beginning of fermentation contributes to the proper cellular structure of yeast [2]. Once properly structured, yeast cells are protected from the increasing alcohol concentration, although in ciders the ethanol concentration is too low to observe this phenomenon. It can be assumed that the use of oxygenation of musts with a higher sugar content would contribute to increase the tolerance of yeast to ethanol. Thus, oxygenated cider would contain higher concentrations of ethanol. Based on the results presented by Petrozziello et al. [20], it can be concluded that the dose of oxygen introduced (7 mg/L, 14 mg/L, 21 mg/L, and 28 mg/L of consumed oxygen) also affects the concentration of ethanol in wines, which decreases with the increase in the amount of oxygen introduced. It was only at a concentration of 28 mg of oxygen/L that a higher concentration of ethanol was observed.

Different relationships were observed for the extract and total acidity. It was found that in the oxygenated ciders (both at the must production and at the fermentation stages), more extract remained in comparison to the control samples. The observed phenomenon was independent of the yeast strain used for fermentation. A larger amount of extract may be caused by an increase in the content of sugar-free compounds that form the extract. This proves that a sufficiently long fermentation process may allow the almost complete use of the available sugars, regardless of the yeast strain [2]. According to research by Satora et al. [21], the presence of oxygen may cause an increase in the content of carbonyl compounds, which is mainly caused by the oxidation of ethanol. The increase in extract concentration may have been associated with quantitative changes in the profile of polyphenols and aroma compounds that occurred after the application of oxygenation.

Sirén et al. [16] observed a slight increase of acidity in wines subjected to oxygenation, similar to the results of this study. Based on research by Petrozziello et al. [20], it can be argued that the concentration of oxygen introduced into a wine has an impact on total acidity. Oxygenation also increases the number of yeast cells, which can lead to an increase in the concentration of byproducts of the fermentation process, such as organic acids. The main secondary end product of alcohol fermentation is succinic acid, but low concentrations of pyruvic, malic, fumaric, oxaloacetic, citric, α-ketoglutaric, glutamic, propionic, and lactic acids are also present [22].

The obtained results for volatile acidity (Table 1) are ambiguous, but the values did not exceed the permissible concentration of 1.3 g acetic acid/L [23]. The volatile acidity of ciders fermented with distillery yeast and wine yeast was similar. The oxygenation of these samples at the must stage did not affect the level of volatile acidity, but oxygenation during fermentation significantly increased it. The control samples fermented with cider yeast were characterized by high volatile acidity, which decreased after oxygenation. Ciders prepared using wild yeast also had high volatile acidity, and oxygenation at the must stage further increased it significantly (Table 1). Research conducted by Du Toit [1] confirmed that oxygenation led to higher acetic acid bacteria counts; however, no increase in volatile acidity was observed in South African Pinotage wine.

Oxygenation slightly affected the use of FAN (Table 1). Significant differences (over 20% compared to the control samples) were noticed only in the case of samples oxygenated in the initial period of fermentation and produced with wild yeast. 

### 3.2. The Effect of Oxygenation on Polyphenol Content and Cider Antioxidant Activity

The polyphenolic profile of apples, and thus cider, depends on apple variety, climate, and degree of ripeness, storage conditions, and processing of the fruit. It is assumed that among the various classes of polyphenols in apples, the highest proportion consists of procyanidins (40–89%). Much lower concentrations of hydroxycinnamic acid, dihydrochalkones, flavonols, anthocyanins, and flavan-3-ols are found [24]. Based on the results of this research and the available literature, it can be concluded that ciders are a source of phenolic compounds, but in comparison with other alcoholic beverages, e.g., wines, they show relatively weak antioxidant properties [24]. 

Oxygenation at the must stage caused a slight increase of antioxidant activity only in the samples fermented with distillery yeast and cider yeasts. The use of oxygenation at the fermentation stage led to a reduction of the antioxidant activity and total polyphenols content, regardless of the yeast strain used (Table 2). This decrease could have been caused by the reaction of quinones (oxidation products of phenolic compounds) with other phenols, resulting in the formation of new larger molecules. These polymers usually have a lower redox potential compared to that of their precursors [25,26]. A decrease of anthocyanin content in oxygenated wines is also observed, and its rate depends on the amount of oxygen supplied [20,27]. Cano-López et al. [28] showed how different wines were differently affected by oxidation. They reported that micro-oxygenated wines had a higher percentage of new anthocyanin-derived pigments, whose formation was more favored in wines with the highest total phenol content. These pigments, in turn, significantly increased wine color intensity [28].

Increases in antioxidant activity and/or polyphenol content were observed in several tests. The first one of these was conducted with fermented samples from either distillery yeast or cider yeast. These samples were oxygenated during the must production stage. A higher concentration of antioxidant activity was found compared to the control samples. Another exception was musts fermented with wine yeast. These showed a higher content of total polyphenolic compounds (TPC) compared with the control. By using oxygenation of the must before and during fermentation, a higher concentration of chlorogenic acid was observed in samples fermented with wine yeast. The last exception was the higher catechin concentration in the samples subjected to oxygenation during fermentation with cider yeast. As a result of oxygenation, the structure of polyphenols may change, for example, the distribution of condensed polyphenols, with the release of chlorogenic acid that synergizes with catechin [29,30]. The exceptions described above concern musts fermented with noble yeast. Oxygenation in fermented musts in the presence of wild yeast resulted in a decrease in the concentration of the analyzed polyphenols. It is possible that noble yeast has a higher resistance to polyphenol oxidation. The use of oxygenation had no effect or caused a decrease in the concentration of caffeic acid and phloridin in all samples analyzed. 

### 3.3. The Effect of Oxygenation on the Profile of Volatile Cider Compounds

Oxidation may improve wine flavor, increasing its fruity aroma and decreasing the vegetal notes. This decrease is not connected with the reduction of typical herbaceous compounds (C6 compounds); however, the reduction of other compounds, e.g., oxidized pyrazines and thiols, may contribute to appearance the herbaceous flavor of wines [31]. As a result of using must oxygenation (Table 3), the content of esters and alcohols in the ciders we analyzed mostly increased. However, due to oxygenation during fermentation, a decrease in the concentration of these volatile components was usually observed. The above relationship may be related to the formation of esters and higher alcohols during fermentation and to their oxidation as a result of the use of oxygenation. The exceptions were isobutyl acetate, 2-methylbutanol, and diethyl acetal, whose concentrations were higher in oxygenated wines compared to the controls. Hernández-Orte et al. [32] observed an increase in the concentration of esters, including ethyl acetate and isobutyl acetate, in Tempranillo wines, which were micro-oxygenated after fermentation. Different results were presented by Du Toit [1], who showed a reduction in isobutyl acetate concentration in South African Pinotage wine after excessive oxidation. Based on research by Perez-Magarino et al. [33], a reduction in the concentration of short-chain esters, acetates, and C6 alcohols was found, as well as an increase in the concentration of succinic acid derivatives and long-chain esters in wine obtained using the combined effect of different chips and the application of oxidation before malolactic fermentation. These changes caused a decrease in green taste, acidity, and bitterness [18].

Oxidation of musts before and during fermentation did not show the same effect on the concentration of terpenes in the samples we analyzed, the content of which increased in some cases (pinocarveol, cider yeast and wild yeast; terpen-4-ol, distillery yeast; geraniol, wine yeast and wild yeast; eugenol, wild yeast; and isoeugenol, cider yeast) and decreased in others (Table 4). A decrease in concentration of eugenol, guaiacol, and isoeugenol, among others, was observed by Hernández-Orte et al. [32] in Tempranillo wines, micro-oxygenated after fermentation. In our research, the concentration of eugenol increased relative to the control when using must oxygenation and decreased when musts were oxygenated during fermentation. Based on research by Ferreira et al. [34], it was discovered that the use of wine oxygenation contributed to an increase of eugenol concentration, which consequently deepened the feeling of a woody aroma. Excessive oxygen also leads to the degradation of β-ionone and β-damascenone, which confer a violet and a ripe fruit aroma, respectively. Also in our research (with two exceptions), a decrease in the concentration of these compounds was observed after the application of oxygenation. Ferreira et al. [35] confirmed a decrease in the levels of terpene alcohols due to the synergistic effect of increasing both temperature and O_2_ content while at low pH levels.

## 4. Conclusions

Oxidation of musts before and during fermentation reduced ethanol concentration in the tested samples. In addition, a higher concentration of extract as well as higher total and volatile acidity levels were observed in the oxygenated samples. The use of oxygenation in most of the studied ciders caused a reduction in polyphenols and oxidative activity, especially in samples fermented with wild yeast. Perhaps, noble yeast has a higher tolerance to oxygen present in must. As a result of using must oxygenation, the content of esters and alcohols in the ciders we analyzed mostly increased. However, due to must oxygenation during fermentation, a decrease in the concentration of these volatile components was usually observed. The above dependence may be related to the formation of esters and higher alcohols during fermentation, which may be oxidized as a result of the use of oxygenation during fermentation; oxygenation can also cause oxidative stress of yeast, which consequently produces fewer aromatic compounds. The oxygenation of the samples did not show the same effect on terpene concentration in the analyzed ciders, the content of which increased in some cases and decreased in others.

## Figures and Tables

**Table 1 biomolecules-10-00890-t001:** Impact of the type of oxygenation on oenological parameters of ciders.

Sample Type	Oxygenation	Ethanol Content [%]	Total Extract [g/L]	Total Acidity [g of malic acid/L]	Volatile Acidity [g of acetic acid/L]	Free Amino Nitrogen (FAN) [mg/L]
DY	control	6.84 ± 0.00 ^a^	16.47 ± 1.33 ^a^	1.44 ± 0.05 ^a^	0.28 ± 0.03 ^a^	30.86 ± 4.99 ^a^
DY	must	5.89 ± 0.05 ^c^	18.60 ± 1.04 ^b^	1.81 ± 0.41 ^b^	0.30 ± 0.06 ^a^	34.92 ± 5.84 ^a^
DY	during fermentation	5.79 ± 0.09 ^c^	18.17 ± 0.99 ^b^	1.79 ± 0.25 ^b^	0.37 ± 0.10 ^b^	25.99 ± 1.57 ^b^
CY	control	6.56 ± 0.20 ^b^	16.03 ± 0.76 ^a^	1.23 ± 0.18 ^c^	0.48 ± 0.13 ^c^	48.82 ± 0.81 ^c^
CY	must	5.87 ± 0.05 ^c^	18.30 ± 0.00 ^b^	1.36 ± 0.12 ^c^	0.29 ± 0.02 ^a^	50.34 ± 2.29 ^c^
CY	during fermentation	5.79 ± 0.04 ^c^	17.83 ± 0.57 ^a,b^	1.47 ± 0.10 ^a^	0.37 ± 0.06 ^b^	40.13 ± 3.34 ^d^
WY	control	6.87 ± 0.05 ^a^	16.57 ± 0.75 ^a^	1.25 ± 0.22 ^c^	0.28 ± 0.05 ^a^	39.93 ± 5.56 ^d^
WY	must	5.84 ± 0.08 ^c^	18.80 ± 1.14 ^b^	1.54 ± 0.05 ^a^	0.28 ± 0.09 ^a^	46.54 ± 5.60 ^c^
WY	during fermentation	5.92 ± 0.00 ^c^	17.90 ± 0.17 ^a,b^	1.55 ± 0.16 ^a^	0.42 ± 0.06 ^b,c^	40.47 ± 5.68 ^d^
WDY	control	6.61 ± 0.00 ^b^	16.43 ± 0.75 ^a^	1.50 ± 0.12 ^a^	0.47 ± 0.08 ^c^	42.29 ± 3.16 ^d^
WDY	must	5.89 ± 0.09 ^c^	18.20 ± 1.45 ^b^	1.81 ± 0.15 ^b^	0.61 ± 0.25 ^d^	41.35 ± 3.44 ^d^
WDY	during fermentation	5.89 ± 0.05 ^c^	18.00 ± 0.00 ^b^	1.63 ± 0.18 ^a,b^	0.49 ± 0.02 ^c^	33.78 ± 1.46 ^a^

DY, distillery yeast; CY, cider yeast; WY, wine yeast, WDY, wild yeast. Means marked with the same letter in a column are not significantly different at *p* < 0.05, n = 3.

**Table 2 biomolecules-10-00890-t002:** Impact of the type of oxygenation on polyphenol content and antioxidant activity of ciders.

Sample Type	Oxygenation	AOX [mg of Trolox/100 mL]	TPC [mg of Catechin/100 mL]	Chlorogenic Acid [mg/L]	Catechin [mg/L]	Phloridzin [mg/L]
DY	control	57.23 ± 2.38 ^a^	12.31 ± 0.52 ^a^	0.24 ± 0.00 ^a^	0.82 ± 0.07 ^a^	0.23 ± 0.08 ^a^
DY	must	63.09 ± 2.17 ^b,d^	11.90 ± 0.60 ^a^	0.19 ± 0.00 ^b^	0.18 ± 0.02 ^b^	-
DY	during fermentation	56.26 ± 1.45 ^a^	11.89 ± 0.13 ^a^	0.20 ± 0.01 ^c^	0.60 ± 0.10 ^c^	-
CY	control	70.34 ± 0.79 ^c^	14.25 ± 0.47 ^b,d^	0.15 ± 0.02 ^d^	0.56 ± 0.08 ^c^	0.24 ± 0.00 ^a^
CY	must	72.27 ± 1.87 ^c^	15.69 ± 0.62 ^c^	0.14 ± 0.00 ^d^	0.31 ± 0.02 ^d^	-
CY	during fermentation	61.53 ± 1.33 ^b^	13.40 ± 0.41 ^b^	0.12 ± 0.00 ^e^	0.79 ± 0.06 ^a^	-
WY	control	66.54 ± 2.80 ^d^	13.40 ± 0.31 ^b^	0.12 ± 0.01 ^e^	0.82 ± 0.03 ^a^	0.15 ± 0.01 ^b^
WY	must	65.07 ± 2.95 ^b,d^	14.82 ± 0.28 ^d^	0.14 ± 0.02 ^d^	0.25 ± 0.02 ^e^	0.10 ± 0.01 ^c^
WY	during fermentation	64.30 ± 1.46 ^d^	12.45 ± 0.45 ^a^	0.15 ± 0.01 ^d^	0.40 ± 0.08 ^d^	-
WDY	control	57.24 ± 1.54 ^a^	11.82 ± 0.46 ^a^	0.10 ± 0.01 ^f^	0.62 ± 0.02 ^c^	0.62 ± 0.02 ^d^
WDY	must	56.61 ± 0.50 ^a^	11.59 ± 0.15 ^a^	0.10 ± 0.01 ^f^	0.22 ± 0.09 ^b^	-
WDY	during fermentation	54.20 ± 2.37 ^a^	10.60 ± 0.68 ^e^	0.10 ± 0.01 ^f^	0.62 ± 0.15 ^c^	-

TPC, total polyphenol content; AOX, antioxidant activity. Means marked with the same letter in a column are not significantly different at *p* < 0.05, n = 3.

**Table 3 biomolecules-10-00890-t003:** Impact of the type of oxygenation on volatile compounds concentration in ciders.

Sample Type	Oxygenation	Ethyl Acetate	Isobutyl Acetate	Isopentyl Acetate	Hexyl Acetate	Ethyl Hexanoate	3-methyl-Butanol	2-methyl-Butanol	Isobutanol	Diethyl Acetal
		mg/L								
DY	C	80.87 ± 10.60 ^a^	1.38 ± 0.04 ^a^	0.48 ± 0.08 ^a^	0.11 ± 0.01 ^a^	0.27 ± 0.05 ^a^	245.82 ± 3.22 ^a^	171.27 ± 2.35 ^a^	169.66 ± 17.53 ^a^	2.31 ± 0.07 ^a^
DY	M	44.27 ± 8.20 ^b^	1.72 ± 0.02 ^b^	1.09 ± 0.24 ^b^	0.15 ± 0.02 ^b,e^	0.31 ± 0.07 ^a,d^	231.01 ± 4.36 ^b^	173.58 ± 5.63 ^a^	159.47 ± 10.45 ^a^	2.83 ± 0.12 ^b,e^
DY	DF	27.13 ± 1.86 ^c^	1.59 ± 0.02 ^c^	0.45 ± 0.01 ^a^	0.08 ± 0.01 ^c^	0.00 ± 0.00 ^b^	246.83 ± 4.06 ^a,b^	178.53 ± 2.60 ^a^	165.86 ± 9.59 ^a^	2.52 ± 0.05 ^c^
CY	C	54.63 ± 3.38 ^d^	1.29 ± 0.04 ^d^	0.37 ± 0.26 ^c^	0.16 ± 0.01 ^b^	0.55 ± 0.04 ^c^	154.05 ± 2.20 ^c^	97.05 ± 2.25 ^b^	58.80 ± 3.64 ^b^	1.92 ± 0.05 ^d^
CY	M	28.26 ± 0.20 ^c^	1.65 ± 0.02 ^e^	0.21± 0.02 ^d^	0.20 ± 0.01 ^d^	0.37 ± 0.00 ^d^	163.49 ± 5.12 ^d^	114.27 ± 4.24 ^c^	76.09 ± 5.36 ^c^	2.77 ± 0.06 ^b^
CY	DF	21.92 ± 1.86 ^e^	1.79 ± 0.01 ^f^	0.11± 0.02 ^e^	0.14 ± 0.01 ^b,e^	0.13 ± 0.01 ^e^	155.91 ± 3.98 ^c^	114.01 ± 2.80 ^c^	73.59 ± 6.81 ^c^	3.00 ± 0.07 ^e^
WY	C	55.86 ± 4.85 ^d^	1.45 ± 0.02 ^a^	0.16 ± 0.01 ^e^	0.13 ± 0.01 ^e^	0.30 ± 0.06 ^a^	154.41 ± 1.34 ^c^	109.49 ± 1.12 ^d^	61.97 ± 2.29 ^b^	2.60 ± 0.10 ^c^
WY	M	32.45 ± 0.93 ^f^	1.76 ± 0.02 ^b^	0.29 ± 0.20 ^d,e^	0.17 ± 0.01 ^b^	0.22 ± 0.05 ^a^	149.60 ± 1.77 ^c^	109.72 ± 1.21 ^d^	68.74 ± 2.36 ^c^	3.49 ± 0.67 ^e,f^
WY	DF	29.94 ± 8.44 ^c,e,f^	1.72 ± 0.03 ^b^	0.08 ± 0.03 ^f^	0.15 ± 0.01 ^b,e^	0.12 ± 0.10 ^e,f^	140.92 ± 3.00 ^e^	109.33 ± 2.73 ^d^	71.48 ± 1.76 ^c^	3.66 ± 0.25 ^e^
WDY	C	41.82 ± 3.98 ^b^	1.52 ± 0.03 ^c^	0.15 ± 0.01 ^e^	0.11 ± 0.01 ^a^	0.18 ± 0.04 ^a,e,f^	139.23 ± 6.85 ^e^	91.34 ± 2.08 ^e^	75.12 ± 3.30 ^c^	2.75 ± 0.10 ^b^
WDY	M	54.81 ± 12.40 ^d^	1.76 ± 0.04 ^f^	0.24 ± 0.12 ^d^	0.13 ± 0.03 ^a,b^	0.16 ± 0.0 ^f^	140.02 ± 1.60 ^e^	98.06 ± 1.19 ^b^	80.92 ± 9.20 ^c^	3.17 ± 0.08 ^e^
WDY	DF	31.55 ± 1.20 ^f^	1.79 ± 0.04 ^f^	0.14 ± 0.03 ^e^	0.11 ± 0.01 ^a^	0.12 ± 0.01 ^e^	156.64 ± 1.68 ^c^	129.72 ± 3.00 ^f^	123.13 ± 4.34 ^d^	3.41 ± 0.11 ^f^

C, control, M, must; DF, during fermentation. Means marked with the same letter in a column are not significantly different at *p* < 0.05, n = 3.

**Table 4 biomolecules-10-00890-t004:** Impact of the type of oxygenation on terpenes concentration in ciders.

Sample Type	Oxygenation	Pinocarveol	Camphor	Terpen-4-ol	Geraniol	Eugenol	β-Damascenone	Isoeugenol	β-Ionone
		mg/L							
DY	C	1.03 ± 0.11 ^a^	0.06 ± 0.01 ^a^	0.54 ± 0.04 ^a^	0.13 ± 0.05 ^a,b^	0.07 ± 0.02 ^a^	0.04 ± 0.00 ^a^	3.83 ± 0.18 ^a,c^	0.09 ± 0.01 ^a^
DY	M	1.03 ± 0.03 ^a^	0.03 ± 0.00 ^b^	1.20 ± 0.04 ^b^	0.14 ± 0.02 ^a^	0.24 ± 0.03 ^b^	0.04 ± 0.00 ^a^	3.34 ± 0.03 ^b^	0.07 ± 0.01 ^a^
DY	DF	0.92 ± 0.02 ^b^	0.03 ± 0.00 ^b^	0.56 ± 0.01 ^a^	0.11 ± 0.00 ^a,b^	0.07 ± 0.02 ^a^	0.04 ± 0.00 ^a^	3.67 ± 0.03 ^a,c^	0.08 ± 0.00 ^a^
CY	C	0.58 ± 0.03 ^c^	0.05 ± 0.00 ^a^	1.95 ± 0.11 ^c^	0.10 ± 0.01 ^b,c^	0.25 ± 0.01 ^b^	0.05 ± 0.00 ^b^	3.64 ± 0.20 ^a,c^	0.08 ± 0.01 ^a^
CY	M	0.60 ± 0.03 ^c^	0.03 ± 0.00 ^b^	1.88 ± 0.08 ^c^	0.09 ± 0.01 ^b,c^	0.33 ± 0.03 ^c^	0.04 ± 0.00 ^a^	3.88 ± 0.21 ^a,c^	0.09 ± 0.01 ^a^
CY	DF	0.68 ± 0.01 ^d^	0.03 ± 0.00 ^b^	0.96 ± 0.06 ^d^	0.08 ± 0.01 ^c^	0.24 ± 0.02 ^b^	0.04 ± 0.00 ^a^	3.66 ± 0.14 ^a,c^	0.08 ± 0.00 ^a^
WY	C	0.67 ± 0.06 ^d^	0.04 ± 0.00 ^b^	1.04 ± 0.03 ^e^	0.08 ± 0.00 ^c^	0.16 ± 0.01 ^c^	0.04 ± 0.00 ^a^	3.42 ± 0.31 ^c^	0.06 ± 0.01 ^b^
WY	M	0.60 ± 0.02 ^c,d^	0.02 ± 0.00 ^c^	1.45 ± 0.05 ^f^	0.09 ± 0.01 ^b,c^	0.28 ± 0.03 ^b^	0.03 ± 0.00 ^c^	3.06 ± 0.13 ^d^	0.07 ± 0.00 ^b^
WY	DF	0.67 ± 0.03 ^d^	0.03 ± 0.00 ^b^	0.81 ± 0.07 ^g^	0.09 ± 0.01 ^b,c^	0.14 ± 0.02 ^d^	0.04 ± 0.00 ^a^	2.39 ± 0.08 ^e^	0.06 ± 0.01 ^b^
WDY	C	0.45 ± 0.04 ^e^	0.03 ± 0.00 ^b^	0.50 ± 0.09 ^a^	0.08 ± 0.00 ^c^	0.10 ± 0.00 ^e^	0.03 ± 0.00 ^c^	3.83 ± 0.25 ^a,c^	0.07 ± 0.02 ^b^
WDY	M	0.55 ± 0.03 ^c^	0.02 ± 0.00 ^c^	0.52 ± 0.11 ^a^	0.08 ± 0.01 ^c^	0.19 ± 0.01 ^f^	0.03 ± 0.00 ^c^	3.74 ± 0.14 ^a^	0.06 ± 0.01 ^b^
WDY	DF	1.04 ± 0.09 ^a^	0.03 ± 0.00 ^b^	0.19 ± 0.04 ^h^	0.09 ± 0.00 ^b,c^	0.12 ± 0.01 ^d^	0.03 ± 0.00 ^c^	4.13± 0.11 ^f^	0.08 ± 0.01 ^a^

Means marked with the same letter in a column are not significantly different at *p* < 0.05, n = 3.

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
