# Peer review of "Effect of Musts Oxygenation at Various Stages of Cider Production on Oenological Parameters, Antioxidant Activity, and Profile of Volatile Cider Compounds"

_biomolecules, 2020, doi:10.3390/biom10060890_

Round 1
Reviewer 1 Report
- Sección 2.1:
- Were the experiences done at least in duplicate? If not, it would be recommended.
- Section 2.6
- According to the reference method (ref. 14: Tarko, T .; Duda-Chodak, A .; Sroka, P .; Satora, P .; Semik-Szczurak, D .; Wajda, Ł. Diversity and bioavailability of fruit polyphenols. J . Food Nutr. Res. 2017, 56, 167-178), "Limit of detection (LOD) and limit of quantification (LOQ) for flavonoids and phenolic acids were 0.02 mg/kg and 0.1 mg/kg, and for anthocyanins 0.05 mg/kg and 0.2 mg/kg respectively”. Section 3.2 Table 2 shows polyphenol data below the LOD and LOQ. It must be corrected
- Section 2.7 and 2.8
- Non-referenced method. Impossible to know the characteristics of the method (LOD, LOQ, LL, etc.).
- The detector used, its characteristics and operational variables are not indicated.
Author Response
See attachment, please.

Reviewer 2 Report
Point 2.1. How many replicates of each experiment were done?
Point 2.4 how many replicates were done for the antioxidant activity of every sample?
Point 2.5 How many replicates for each sample were made for the determination of total polyphenol content?
Line 65 “Cracow” ? Do you mean Krakow? If yes please correct.
Line 70 “Some of the grafted samples” it is better to explain that for each yeast there was a must oxygenation, oxygenation during fermentation and a control.
Line 74 put in please small letters oxygenation during fermentations
Line 96 please mention that total polyphenols were expressed as catechin equivalent.
Point 2.7 The authors need to add more information on this point. How were the volatile compounds Identified? Did you use a library? Which one? What type of chromatograph was used? What type of detector was used?
Point 2.8. The terpenoids are also volatile compounds, why they were not identified together with the rest of the compounds using the SPME extraction method?. Here again misses the information: How were the volatile compounds Identified? Did you use a library? Which one? What type of chromatograph was used? What type of detector was used?
Point 3.1 please make a comment on this what is the expected ethanol concentrations in ciders.
Are the lower concentrations of ethanol in the expected range?
Line 179 “This proves that a sufficiently long fermentation process may allow almost complete
use of available sugars, regardless of the yeast strains” Does this mean that the experiments without oxygenation were longer in time? If yes please put this in the manuscript.
Line 180 “Higher concentration of the extract may be caused by an increase in the content of compounds forming sugar-free extract.” What does this mean? What are these compounds forming sugar-free extract? May be I did not understood the phrase. Is there a comer after “compounds”? If the phrase is : “Higher concentration of the extract may be caused by an increase in the content of compounds, forming sugar-free extract.” With the comer this has different sense.
Line 193 “The use of must and cider oxygenation at the initial stage of fermentation also increased volatile acidity” This is only true for the DY yeast. In general the results about the volatile acidity are inconclusive. For the different yeasts the results were different. For WY yeasts there were not differences in terms of VA between control and must. And for yeast WDY there were not differences
Between control and oxidation during fermentation. For the four yeast here we have all possible results. Please re-write this paragraph.
Line 218 “decrease in the concentration of ethanol in the polyphenols” I do not understand this. Please rewrite.
Line 224 “The higher the polyphenol content, the better the benefits of oxygenation” I don’t understand this statement, as oxygenation is bad for polyphenols. The benefit is for what?
Line 226 “The formation of anthocyanin-derived pigments was more intense in wines with high total phenol content, especially those with the highest quantities of free anthocyanins which obtained the best results” This information is unclear for me. The formation of anthocyanin-derived pigments were on the base of what? Why are they formed? Were obtained best results for what? Best results after oxygenation? Please rewrite.
Line 228-231 The said in these lines is not true as according to table 2 the antioxidant activity of samples with cyder yeast was not higher than the control. Please rewrite this.
Line 233 “By using oxygenation of the must before and during fermentation, a higher concentration of chlorogenic acid was observed in fermented samples.” This is also not true. This can be said only for WY samples.
Lines 228-235 This text in general is difficult to follow. It can be re-written.
In tables 3 and 4 the values of standard deviation are on the row below. Please correct.
Author Response
See attachment, please.
Round 2
Reviewer 1 Report
Line 169-171 In the text: "Because the concentration of tested compounds was lower than LOQ, the samples were previously concentrated by extraction with hexane." Has the extraction stage been studied? It is not described, how is it? What are the concentration factors for the compounds? This information must be included.
Author Response
Line 169-171 In the text: "Because the concentration of tested compounds was lower than LOQ, the samples were previously concentrated by extraction with hexane." Has the extraction stage been studied? It is not described, how is it? What are the concentration factors for the compounds? This information must be included.
Thank you very much for the above remark. The wording in the manuscript was misleading and was deleted. The LOD and LOQ values given in the original version were calculated for a sample extracted with hexane in a ratio of 4 mL sample and 4 mL hexane. Because the concentrations of terpenoids in the ciders were lower, extraction was finally used in a ratio of 40 (sample) : 4 (hexane). LOD and LOQ values are 0.003 mg / L and 0.01 mg / L, respectively. The authors optimized the extraction method using various extractants, ratio of volume sample to solvent, and rotary shaker parameters. An internal standard was used to quantify terpenoids. The developed method was also validated by analyzing terpenoids of known concentration. The appropriate correction was introduced to the content of the manuscript (line168-169).
Reviewer 2 Report
I have no more comments.
Author Response
Thank you very much